# 3′,8″-Dimerization Enhances the Antioxidant Capacity of Flavonoids: Evidence from Acacetin and Isoginkgetin

**DOI:** 10.3390/molecules24112039

**Published:** 2019-05-28

**Authors:** Xican Li, Xiaojian Ouyang, Rongxin Cai, Dongfeng Chen

**Affiliations:** 1School of Chinese Herbal Medicine, Guangzhou University of Chinese Medicine, Waihuan East Road No. 232, Guangzhou Higher Education Mega Center, Guangzhou 510006, China; oyxiaojian55@163.com (X.O.); choi_roy@foxmail.com (R.C.); 2Innovative Research & Development Laboratory of TCM, Guangzhou University of Chinese Medicine, Waihuan East Road No. 232, Guangzhou Higher Education Mega Center, Guangzhou 510006, China; 3School of Basic Medical Science & Research Center of Basic Integrative Medicine, Guangzhou University of Chinese Medicine, Guangzhou 510006, China; 4The Research Center of Basic Integrative Medicine, Guangzhou University of Chinese Medicine, Guangzhou 510006, China

**Keywords:** 3′,8″-dimerization, acacetin, antioxidant, isoginkgetin, radical adduct formation

## Abstract

To probe the effect of 3′,8″-dimerization on antioxidant flavonoids, acacetin and its 3′,8″-dimer isoginkgetin were comparatively analyzed using three antioxidant assays, namely, the **·**O_2_^−^ scavenging assay, the Cu^2+^ reducing assay, and the 2,2′-azino bis(3-ethylbenzothiazolin-6-sulfonic acid) radical scavenging assay. In these assays, acacetin had consistently higher IC_50_ values than isoginkgetin. Subsequently, the acacetin was incubated with 4-methoxy-2,2,6,6-tetramethylpiperidine-1-oxy radicals (4-methoxy-TEMPO) and then analyzed by ultra-performance liquid chromatography coupled with electrospray ionization quadrupole time-of-flight tandem mass spectrometry (UHPLC−ESI−Q−TOF−MS) technology. The results of the UHPLC−ESI−Q−TOF−MS analysis suggested the presence of a dimer with *m*/*z* 565, 550, 413, 389, 374, 345, 330, and 283 peaks. By comparison, standard isoginkgetin yielded peaks at *m*/*z* 565, 533, 518, 489, 401, 389, 374, and 151 in the mass spectra. Based on these experimental data, MS interpretation, and the relevant literature, we concluded that isoginkgetin had higher electron transfer potential than its monomer because of the 3′,8″-dimerization. Additionally, acacetin can produce a dimer during its antioxidant process; however, the dimer is not isoginkgetin.

## 1. Introduction

Flavonoids are an important source of natural phenolic antioxidants [1]. Thousands of flavonoids have been documented from medicinal plants, especially from Chinese herbal medicines (CHMs) [2,3]. These flavonoids can be classified into two types: monoflavonoids (e.g., quercitrin [4]) and polyflavonoids (e.g., tetrahydroamentoflavone [5]). The so-called polyflavonoids, however, are predominantly biflavonoids, and biflavonoids can be found to co-exist with monoflavonoids in a same plant (especially CHMs) [6]. By comparison, triflavonoids and tetraflavonoids are very rare in natural products [2,7,8,9]. 

Natural biflavonoids can be classified into five main subtypes: 3′,8″-linkage biflavonoids, 3,8″-linkage biflavonoids, 3′,6″-linkage biflavonoids, 4′-*O*,6″-linkage biflavonoids, and 4′-*O*,3‴-linkage biflavonoids (Appendix A) [5,7,10,11,12]. Among these, 3′,8″-linkage biflavonoids are the most common and have been specifically termed as 3’,8’’-biflavonoids in some literature reports [13]. Some 3′,8″-biflavonoids, however, comprise of two equivalent monoflavone residues; thus, they can be regarded as the dimerization product of the monoflavone. For example, isoginkgetin can be considered as the 3′,8″-dimerization product of acacetin. However, it is unclear whether there is a difference in antioxidant activity between the monoflavone and its 3′,8″-dimer, and how 3′,8″-dimerization affects the antioxidant capacity of flavonoids.

To address these questions, isoginkgetin and its monomer acacetin were selected as the model compounds in this study. As seen in Figure 1, acacetin is 5,7-dihydroxy-4′-methoxyflavone. Isoginkgetin links two acacetin residues via the 3′,8″-sites; thus, it can be considered as the 3′,8″-dimer of acacetin. Thereby, it is believed that the comparison between isoginkgetin and acacetin will provide important evidence as to whether there is a difference in antioxidant potential between a monoflavone and its 3′,8″-dimer. 

In addition to isoginkgetin, there are at least six 3′,8″-biflavonoids, such as ginkgetin, bilobetin, sciadopitysin, amentoflavone, tetrahydroamentoflavone, and amentoflavone-4′,4‴,7,7″-tetramethyl ether (Appendix A). However, some of these 3′,8″-biflavonoids are constructed with two nonequivalent flavonoid monomers. For example, bilobetin is made up of an acacetin residue and an apigenin residue. Such a difference may bring about some uncertainties for the comparison. The other 3′,8″-biflavonoids (or their flavonoid monomers) are, however, unavailable. In fact, the flavonoid monomer of amentoflavone-4′,4‴,7,7″-tetramethyl ether (i.e., amentoflavone-4′,7-dimethyl ether) has not yet been found. Thus, isoginkgetin and acacetin can be considered as an ideal pair for the comparative study. 

In the present study, isoginkgetin and acacetin were first comparatively investigated for their superoxide anion radical (**·**O_2_^–^) scavenging effects. **·**O_2_^–^ is an important form of reactive oxygen species (ROS) [14,15,16], and a comparison based on the **·**O_2_^–^ scavenging assay can reflect the ROS scavenging levels. Subsequently, isoginkgetin and acacetin were compared using the Cu^2+^ reducing antioxidant assay and the 2,2′-azino bis(3-ethylbenzothiazolin-6-sulfonic acid) (ABTS^+^**·**) scavenging antioxidant assay. Both the Cu^2+^ reducing assay and the ABTS^+^**·** scavenging assay involved electron-transfer (ET) mechanisms and hence may give experimental evidence for the effect of 3′,8″-dimerization on the ET potential of antioxidant flavonoids. Finally, acacetin was mixed with a stable oxygen-centered radical, the 4-methoxy-2,2,6,6-tetramethylpiperidine-1-oxy radical (4-methoxy-TEMPO), and then analyzed by using cutting-edge ultra-performance liquid chromatography coupled with electrospray ionization quadrupole time-of-flight tandem mass spectrometry (UHPLC−ESI−Q−TOF−MS) to test the possibility of acacetin conversion into isoginkgetin.

Undoubtedly, the evidence from isoginkgetin and acacetin can help us understand the antioxidant action of other 3′,8″-biflavones such as amentoflavone, tetrahydroamentoflavone, and amentoflavone-4′,4‴,7,7″-tetramethyl ether [5,7,10,11]. Such evidence may also aid medicinal chemists in designing novel and effective antioxidant molecules.

## 2. Results and Discussion

As important natural antioxidants, both monoflavonoids and biflavonoids effectively scavenge ROS. To compare their relative ROS scavenging levels, acacetin and its 3′,8″-dimer isoginkgetin were examined by using the **·**O_2_^–^ scavenging assay. This assay is based on the pyrogallol autooxidation principle [17] and was improved by our team in a previous study [18]. The improved pyrogallol assay demonstrated that the **·**O_2_^–^ scavenging percentages of both the flavonoids were increased in a dose-dependent manner (Appendix A); however, acacetin had a higher IC_50_ value than that of isoginkgetin (Table 1). This means that both flavonoids can act as an antioxidant to effectively scavenge **·**O_2_^–^, a typical ROS. However, the higher IC_50_ value means that acacetin is an inferior antioxidant to isoginkgetin.

The mechanism of **·**O_2_^–^ scavenging has been reported to involve ET [19]. To test the possibility of ET in this system, both flavonoids were comparatively examined using the Cu^2+^ reducing assay in a pH 7.4 aqueous buffer. The results indicated that the amount of Cu^2+^ reduced by acacetin and isoginkgetin increased in a dose-dependent manner (Appendix A), implying that these flavonoids possessed potential ET activity in a pH 7.4 aqueous solution. However, acacetin possessed a higher IC_50_ value than isoginkgetin (Table 1).

Cu^2+^ is not a free radical; however, the Cu^2+^ reducing assay is similar to a radical scavenging assay. To fully explore their ET potential in radical scavenging, the two flavonoids were further measured by using the ABTS^+^**·** scavenging assay. The ABTS^+^**·** solution was prepared by an ET reaction (Section 3.4), where (NH_4_)_2_ABTS was oxidized by K_2_S_2_O_8_ to give ABTS^+^**·**, a radical with a λ_734 nm_ absorption peak and dark-green appearance (Appendix A) [20]. Thus, the ABTS^+^**·** scavenging assay is considered as an ET-based radical scavenging reaction [5,21]. In the assay reaction, the (phenolic) antioxidant transfers one electron to ABTS^+^**·**, which diminishes the λ_734 nm_ absorption (Appendix A).

The ABTS^+^**·** scavenging assay suggested that both acacetin and isoginkgetin effectively increased their ABTS^+^**·** scavenging ability with increasing concentration (Appendix A), indicating that both flavonoids might employ an ET pathway to exert their antioxidant action. Nevertheless, the ET potential of acacetin was weaker than that of isoginkgetin, according to the IC_50_ values listed in Table 1. Combined with the Cu^2+^ reducing assay results, it can be deduced that acacetin is indeed inferior to isoginkgetin in an antioxidant capacity through ET, and that 3′,8″-dimerization can improve the ET potential of antioxidant flavonoids.

As seen in Figure 1C,D and Appendix A, isoginkgetin contains two planar acacetin residues; however, the two planar moieties do not share the same plane. There is a dihedral angle of 45° between the two acacetin planes. A similar dihedral angle can also be observed in other 3′,8″-biflavonoids, such as (+)-morelloflavone [13] and amentoflavonoid [22]. The dihedral angle of 45° implies that the two acacetin residues are partially planar, and thus, partial π–π conjugation can occur through the 3′,8″-linkage. This partial π–π conjugation can slightly increase the ET potential, as seen in the *Z*-resveratrol molecule [23]. Thus, isoginkgetin possesses higher ET potential than acacetin. The ET antioxidant potential is known to be associated with the cytoprotective effect. Thus, our findings can be used to explain why 3′,8″-biflavone amentoflavonoid possesses better chemoprevention and neuroprotective effects than its mono-flavonoid apigenin [24,25,26].

Previously, our group reported that phytophenols may generate dimers after incubation with free radicals [27]. To verify whether acacetin can give a dimer, it was treated with 4-methoxy-TEMPO for 24 h in this study. The UHPLC−ESI−Q−TOF−MS determination revealed that acacetin yielded a small chromatographic peak at a retention time of 1.8269 min (Figure 2D). This chromatographic peak was associated with a MS peak with *m*/*z* 566, a value of twice the molecular weight of acacetin (Figure 2E), implying the generation of one dimer of acacetin. This dimer further produced several MS/MS peaks, including those at *m*/*z* 565, 550, 413, 389, 374, 345, 330, and 283 (Figure 2F). In accordance with these MS spectra, this dimer was elucidated as that shown in Figure 3A. It should be emphasized that other possible linkages between the two acacetin residues cannot be excluded, such as 7-O, 5″-O, 6-C, 8-C, and 6″-C. If a C–O linkage is present, one of the phenolic -OH groups may be involved in the dimerization; thus, the dimeric product is presumed to be less active than the mono-flavonoid. Of course, the identification of the linkage will be the subject of further research in the future.

Suffice to say, this dimer is not identical to isoginkgetin. As seen in Figure 2I, the standard isoginkgetin yielded eight MS/MS peaks, including those at *m*/*z* 565, 533, 518, 489, 401, 389, 374, and 151. These MS/MS peaks can be elucidated, as shown in Figure 3B. The elucidation further suggested that acacetin does not produce isoginkgetin upon interaction with the methoxy-TEMPO radical. Nevertheless, the *m/z* values with high resolution suggested that there were several same moieties between the dimeric acacetin molecule and the isoginkgetin molecule. The typical *m/z* values included 565.1132, 389.0560, and 374.0416 in dimeric acacetin (Figure 2F), and 565.1137, 389.0647, and 374.0412 in isoginkgetin (Figure 2I). The relative deviations of these corresponding *m/z* values were calculated as 8.8 × 10^−7^, 2.2 × 10^−5^, and 1.0 × 10^−6^, respectively. 

## 3. Materials and Methods

### 3.1. Chemicals

Acacetin (C_16_H_12_O_5_, CAS number 480-44-4, MW 284.2, purity 98%; Appendix A) and isoginkgetin (C_32_H_22_O_10_, CAS number 548-19-6, MW 566.5, purity 98%; Appendix A) were obtained from Chengdu Biopurify Phytochemicals, Ltd. (Chengdu, China). 4-Methoxy-TEMPO**·** (C_18_H_12_N_5_O_6_, CAS number 95407-69-5) was purchased from TCI Shanghai Ltd. (Shanghai, China). 2,9-Dimethyl-1,10-phenanthroline (neocuproine), pyrogallol (2,3-dihydroxyphenol), and (±)-6-hydroxyl-2,5,7,8-tetramethlychromane-2-carboxylic acid (Trolox) were obtained from Sigma-Aldrich (Shanghai, China). (NH_4_)_2_ABTS [2,2′-azino-bis (3-ethylbenzo-thiazoline-6-sulfonic acid diammonium salt)] was obtained from the Amresco Chemical Co. (Solon, OH, USA). Tris-hydroxymethyl amino methane (Tris) was obtained from Dingguo Biotechnology, Ltd (Beijing, China). Water, methanol, and acetonitrile were of HPLC grade and the other reagents were purchased as analytical grade from Guangdong Guanghua Chemical Plants Co. Ltd (Shantou, China).

### 3.2. Superoxide Anion Radical (**·**O_2_^–^) Scavenging Assay

The superoxide anion radical (**·**O_2_^–^) scavenging activity was determined using the improved pyrogallol autooxidation method [18]. In brief, the sample was dissolved in methanol (1 mg/mL). The sample solution (*x* = 5–25 μL) was mixed with Tris-HCl buffer (980 − *x* μL, 0.05 M, pH 7.4) containing Na_2_EDTA (1 mM). Pyrogallol (20 μL, 60 mM in 1 mM HCl) was added, and the mixture was thoroughly shaken at room temperature for 3 s. The absorbance of the mixture was measured (Unico 2100, Shanghai, China) at 325 nm every 30 s for 5 min. Tris-HCl buffer was applied as a blank. The **·**O_2_^–^ inhibiting ability was calculated as follows: (1)$$\mathrm{Inhibition}\;\%=\frac{\left(\frac{\Delta A_{325\mathrm{nm},\mathrm{control}}}{T}-\frac{\Delta A_{325\mathrm{nm},\mathrm{sample}}}{T}\right)}{\left(\frac{\Delta A_{325\mathrm{nm},\mathrm{control}}}{T}\;\right)}\times 100\%$$
where Δ*A*_325nm, control_ is the increment in the absorbance at 325 nm of the mixture without the sample and Δ*A*_325nm, sample_ is the increment of the mixture with the sample; *T* = 5 min.

### 3.3. Cupric Ion (Cu^2+^) Reducing Antioxidant Capacity (CUPRAC) Assay

The cupric ion reducing antioxidant capacity (CUPRAC) assay was determined based on the method proposed by Apak [28], with small modifications, as presented by Tian [29]. Twelve microliters of CuSO_4_ solution (0.01 M) and 12 μL of ethanolic neocuproine solution (7.5 mM) were added to a 96-well plate and mixed with different concentrations of samples (4–20 μg/mL). The total volume was then adjusted to 100 μL with a CH_3_COONH_4_ buffer solution (0.1 M), and mixed again to homogenize the solution. The mixture was maintained at room temperature for 30 min, and the absorbance was measured at 450 nm on a microplate reader (Multiskan FC, Thermo Scientific, Shanghai, China). The relative reducing power of the sample was calculated as follows:(2)$$Relative\;reducing\;power\%=\frac{A-A_{min}}{A_{max}-A_{min}}\times 100\%$$
where *A_max_* is the maximum absorbance, *A_min_* is the minimum absorbance, and *A* is the absorbance of the sample.

### 3.4. ABTS^+^**·** Cation Radical Scavenging Assay

The ABTS^+^**·** cation radical was generated from the reaction of (NH_4_)_2_ABTS aqueous solution (7.4 mM) with K_2_S_2_O_8_ aqueous solution (2.6 mM). In brief, the (NH_4_)_2_ABTS aqueous solution was mixed with the K_2_S_2_O_8_ aqueous solution at room temperature. After an incubation of 12 h, the resulting mixture became dark green. Then, it was diluted about 50 times with methanol. The diluted solution was scanned by using a UV–Vis spectrophotometer (Unico 2600A, Shanghai, China) from 200 to 900 nm and used as the working solution for this assay [30].

To the above ABTS^+^**·** working solution (80 μL), the test sample (*x* = 0–10 μL, 1 mg/mL) and methanol (20 − *x* μL) were added to adjust the total reaction system to 100 μL. After incubating for 6 min, the absorbance of the 100 μL mixture was measured at 734 nm by using a microplate reader (Multiskan FC, Thermo Scientific, Shanghai, China). The percentage inhibition was calculated based on the formula as follows:(3)$$\mathrm{Inhibition}\;\%=\frac{A_{0}-A}{A_{0}}\times \;100\%$$
where *A*_0_ is the absorbance of the control without the sample and *A* is the absorbance of the reaction mixture with the sample.

### 3.5. UHPLC−ESI−Q−TOF−MS Analysis

The reaction proceeded under the conditions described in previous reports [31,32]. In brief, a 0.05 mL methanol solution of acacetin (10.6 mM) was mixed with a 0.05 mL methanol solution of 4-methoxy-TEMPO (21.2 mM). The molar ratio of acacetin to 4-methoxy-TEMPO was 1:2. This reaction mixture was incubated for 10 h at room temperature and then diluted to 2 mL with methanol then filtered through a 0.22-μm filter. The filtrate (2 mL) was analyzed by using a UHPLC−ESI−Q−TOF−MS system (AB SCIEX, Framingham, MA, USA).

The UHPLC−ESI−Q−TOF−MS analysis apparatus was equipped with a Phenomenex Luna C_18_ column (2.1 mm i.d. × 100 mm, 1.6 μm, Phenomenex Inc., Torrance, CA, USA). The mobile phase in the chromatography was made up of a mixture of acetonitrile (phase A) and 0.1% formic acid in water (phase B). The column was eluted at a flow rate of 0.2 mL/min with the following gradient elution program: 0–2 min, 30% B; 2–10 min, gradient decreasing from 30% to 0% B; 10–12 min, gradient increasing from 0% to 30% B. The sample injection volume was 5 μL. The Q–TOF–MS analysis was performed under the negative ionization mode in a Triple TOF 5600*^plus^* mass spectrometer (AB SCIEX, Framingham, MA, USA) equipped with an ESI source. The scan range was set at 100–2000 Da. The system was run with the following parameters: Ion spray voltage, −4500 V; ion source heater, 550 °C; curtain gas (CUR, N_2_), 30 psi; nebulizing gas (GS1, air), 50 psi; and Tis gas (GS2, air), 50 psi. The declustering potential (DP) was set at −100 V, whilst the collision energy (CE) was −45 V with a collision energy spread (CES) of 10 V. For comparison, the standard isoginkgetin solution (0.09 mM, 1 mL) was also analyzed using the above UHPLC–ESI–Q–TOF–MS conditions.

### 3.6. Statistical Analysis

The concentration response curves were analyzed using Origin 6.0 professional software (OriginLab, Northampton, MA, USA). The IC_50_ value was defined as the final concentration required for 50% radical inhibition (or relative reducing power) [33]. It was calculated by linear regression analysis and expressed as the mean ± standard deviation (*n* = 3). The linear regression was analyzed using Origin 6.0. The determination of significant differences between the mean IC_50_ values was performed by using one-way ANOVA analysis and the *t*-test. The analysis was performed using SPSS software 13.0 (SPSS Inc., Chicago, IL, USA) for Windows. *p* < 0.05 was considered statistically significant.

## 4. Conclusions

Acacetin and its 3′,8″-dimer isoginkgetin may mediate ET to exert antioxidant action. However, when compared with acacetin, isoginkgetin has an enhanced ET antioxidant capacity; this enhancement may be attributed to the 3′,8″-dimerization, which allows partial π–π conjugation. Acacetin can produce a dimer during the antioxidant process; however, this dimer is not isoginkgetin. 

## Figures and Tables

**Figure 1 molecules-24-02039-f001:**
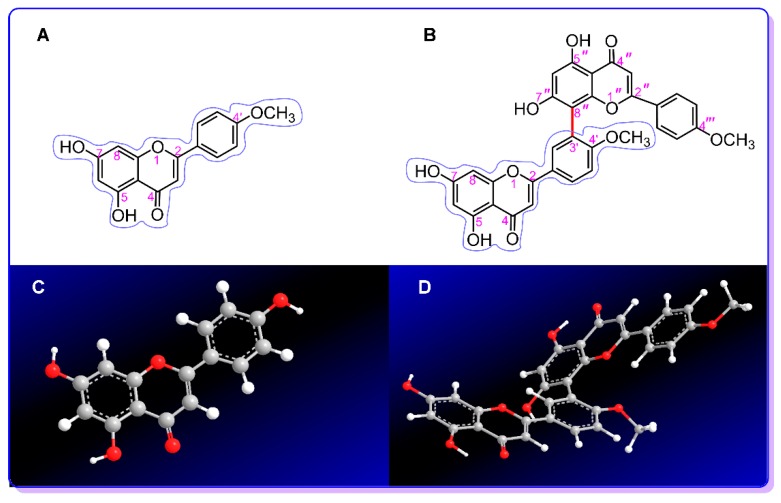
The structures and molecular models of acacetin and isoginkgetin: (**A**) structure of acacetin; (**B**) structure of isoginkgetin; (**C**) molecular model of acacetin; (**D**) molecular model of isoginkgetin. The molecular model was created based on the preferential conformation by using the Chem3D Pro 14.0 program (PerkinElmer, Waltham, MA, USA). The molecular model D has a dihedral angle of 45°, see Appendix A.

**Figure 2 molecules-24-02039-f002:**
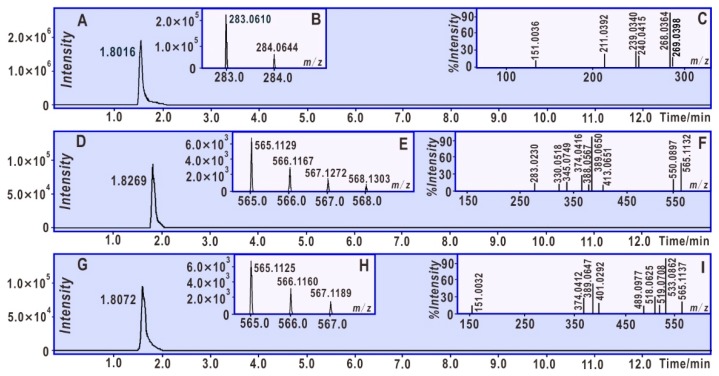
Main results of the ultra-performance liquid chromatography coupled with electrospray ionization quadrupole time-of-flight tandem mass spectrometry (UHPLC−ESI−Q−TOF−MS) analysis of acacetin and isoginkgetin mixed with 4-methoxy-TEMPO radicals (negative ion model). (**A**) Chromatographic peaks of acacetin when the formula [C_16_H_12_O_5_-H]^−^ was extracted; (**B**) MS spectra of acacetin; (**C**) MS/MS spectra of acacetin (from the precursor *m*/*z* 283.0610); (**D**) chromatographic peaks of dimeric acacetin when the formula [C_32_H_22_O_10_-H]^−^ was extracted; (**E**) MS spectra of dimeric acacetin; (**F**) MS/MS spectra of dimeric acacetin (from the precursor *m*/*z* 565.1129); (**G**) chromatographic peaks of authentic isoginkgetin when the formula [C_32_H_22_O_10_-H]^−^ was extracted; (**H**) MS spectra of authentic isoginkgetin; and (**I**) MS/MS spectra of authentic isoginkgetin (from the precursor *m*/*z* 565.1125).

**Figure 3 molecules-24-02039-f003:**
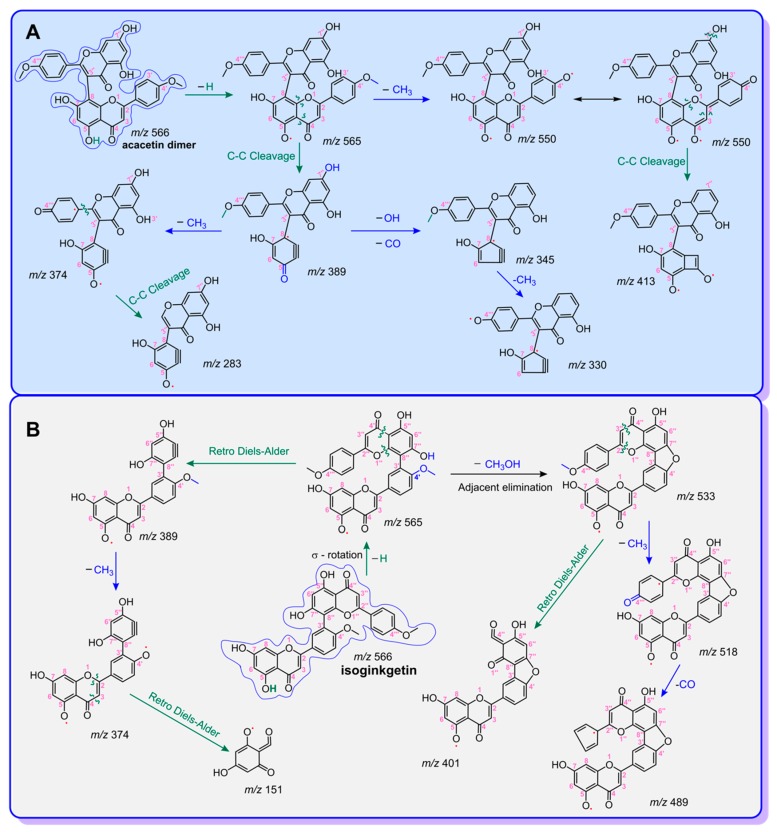
Proposed MS elucidation of acacetin dimer (**A**) and isoginkgetin (**B**).

**Table 1 molecules-24-02039-t001:** IC_50_ values of acacetin and isoginkgetin in various antioxidant assays (μM).

Antioxidant Assays	Acacetin	Isoginkgetin	Trolox
**·**O_2_^–^ scavenging	107.1 ± 11.3 ^b^	33.1 ± 0.4 ^a^	4968.4 ± 157.9
Cu^2+^ reducing	1506.2 ± 33.4 ^b^	367.1 ± 21.4 ^a^	38.1 ± 1.7
ABTS^+^**·** scavenging	246.2 ± 11.8 ^b^	209.3 ± 15.1 ^a^	22.6 ± 0.3

The IC_50_ value is defined as the lowest concentration with 50% radical inhibition or relative reducing power, calculated by linear regression analysis, and expressed as the mean ± SD (*n* = 3). The linear regression was analyzed by using Origin 6.0 professional software. The IC_50_ values with different superscripts (^a^ and ^b^) between acacetin and isoginkgetin are significantly different (*p* < 0.05). Trolox was used as the positive control. All dose-dependent curves are given in Appendix A.

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
