# Peer review of "3′,8″-Dimerization Enhances the Antioxidant Capacity of Flavonoids: Evidence from Acacetin and Isoginkgetin"

_molecules, 2019, doi:10.3390/molecules24112039_

Round 1

Reviewer 1 Report

The manuscript presented by Li and collaborators tells a story on bioflavonoids' antioxidant properties. 

It needs to be written better, once the reader doesn't know why it is important to compare the activities of acacetin and isoginkgetin. 

Just by comparing the structures of these two compounds, it is evident that the isoginkgetin would be more active. 

Also, it is recommended to present the results in a different manner, start with theoretical predictions, MS/MS data and finish with the activity tests. 

It is also recommended to organize all antioxidant results for the aforementioned compounds in one Table and discuss them as – what experimental data on other bioflavonoids tell on the activity properties of these two compounds?

Do not use Q-TOF and MS/MS in the same abbreviation as Q-TOF is a MS/MS system.

Reviewer 2 Report

The manuscript molecules-496632, titled  "3’,8”-Dimerization Enhances the Antioxidant Capacity of Flavonoids: Evidence from Acacetin and Isoginkgetin” reports the antioxidative activity of two natural products belonging to the family of flavonoids.  Precisely the autors use the flavone acacetin and its 3’,8”-dimer to spectroscopically measure the antioxidant activity by means of three different methods.

This work is poor of contents, poor of experiments and it does not match nor the Journal criteria either the aim of the Special Issue “The Antioxidant Capacities of Natural Products 2019".

In fact, “this Special Issue may include original research articles and reviews on new extraction procedures; isolation, purification, and characterization of new compounds; in vitro and in vivo studies on the antioxidant properties of extracts, fractions, synergistic mixtures, and isolated compounds and their possible employment to treat human diseases. Studies dealing with new formulations containing antioxidants (including polymers for active packaging) are also welcome”.

The present work has not novelty object. It  does not deal with extraction procedures or characterization of new natural products. Conversely the authors employed well-known antioxidative assay on two commercially available molecules

The authors try to explain the results observed based on a molecular modelling corresponding to a minimization energy obtained by Chem3D which is not a common software employed in molecular modelling.  

Also the experiment carried out by UHPLC-ESI-MS is poor and incomplete.

This work in the present form can not be accepted as research article.

Reviewer 3 Report

Manuscript is well written but some minor corrections have to be done

Page 2, line 63 – correct references

Correct reference sin text and reference list according to author guidelines

Some signs are left in red

Reviewer 4 Report

The paper is dealing with the dimerization effect on the increase in the antioxidant activity of flavonoids. Manuscript is well written but before publication still small modification are needed.

Please complete the conclusions with practical application.

Round 2

Reviewer 1 Report

Although revised manuscript submitted by Li et al. has been improved in comparison to the first manuscript version, some important corrections must be introduced. Conclusions are very short and do not explain the main results of the presented research. This manuscript section must give answers to the aims and hypothesis that were given in the introduction section. It is also important to compare the ox reductive potential of the studied compounds and explain – why one should use dimers instead of monomer bioflavones?   

Reviewer 2 Report

The present form of the manuscript submitted by Li et al. has been undoubtedly improved respect to the previously one. I still think this work misses in some experiments and it does not fit with the special issue aim according to what reported in the journal home page (as well as I previously copied). However, if the editor and other reviewers believe that the manuscript can be published in this form, I have no further objection.
